# Investigating the Role of Non-Coding RNA in Non-Alcoholic Fatty Liver Disease

**DOI:** 10.3390/ncrna10010010

**Published:** 2024-01-31

**Authors:** Samar A. Zailaie, Basmah B. Khoja, Jumana J. Siddiqui, Mohammad H. Mawardi, Emily Heaphy, Amjad Aljagthmi, Consolato M. Sergi

**Affiliations:** 1Research Center, King Faisal Specialist Hospital & Research Center-Jeddah (KFSHRC-J), Jeddah 21499, Saudi Arabia; zsamar@kfshrc.edu.sa (S.A.Z.); basma.khoja@hotmail.com (B.B.K.); eheaphy@kfshrc.edu.sa (E.H.); aaljagthmi@kfshrc.edu.sa (A.A.); 2Biochemistry Department, Faculty of Science, King Abdulaziz University, Jeddah 21589, Saudi Arabia; jumana.j.siddiqui@gmail.com; 3Medicine Department, Gastroenterology Section, King Faisal Specialist Hospital & Research Center-Jeddah (KFSHRC-J), Jeddah 21499, Saudi Arabia; mmawardi@kfshrc.edu.sa; 4Children’s Hospital of Eastern Ontario (CHEO), University of Ottawa, Ottawa, ON K1H 8L1, Canada; 5Department of Laboratory Medicine and Pathology, University of Alberta, Edmonton, AB T6G 2R3, Canada

**Keywords:** ncRNA, miRNA, PiwiRNA, LncRNA, CircRNA, NAFLD, NASH, oxidative stress, metabolic dysfunction, insulin resistance

## Abstract

Non-coding RNAs (ncRNAs) are RNA molecules that do not code for protein but play key roles in regulating cellular processes. NcRNAs globally affect gene expression in diverse physiological and pathological contexts. Functionally important ncRNAs act in chromatin modifications, in mRNA stabilization and translation, and in regulation of various signaling pathways. Non-alcoholic fatty liver disease (NAFLD) is a set of conditions caused by the accumulation of triacylglycerol in the liver. Studies of ncRNA in NAFLD are limited but have demonstrated that ncRNAs play a critical role in the pathogenesis of NAFLD. In this review, we summarize NAFLD’s pathogenesis and clinical features, discuss current treatment options, and review the involvement of ncRNAs as regulatory molecules in NAFLD and its progression to non-alcoholic steatohepatitis (NASH). In addition, we highlight signaling pathways dysregulated in NAFLD and review their crosstalk with ncRNAs. Having a thorough understanding of the disease process’s molecular mechanisms will facilitate development of highly effective diagnostic and therapeutic treatments. Such insights can also inform preventive strategies to minimize the disease’s future development.

## 1. Introduction

Non-alcoholic fatty liver disease (NAFLD) is a global health issue, particularly in children who have adopted a sedentary lifestyle and received home or virtual schooling. NAFLD is characterized by excess fat in the liver cells, known as hepatocytes. The chronic stress caused by lipid accumulation within the liver may lead to non-alcoholic steatohepatitis (NASH), an inflammatory type of fatty liver. NASH is considered a risk factor for liver fibrosis, cirrhosis, and eventual hepatic carcinogenesis [1]. The prevalence of NAFLD is increasing over time with an estimated global rate of about 38% [2]. The etiology behind NAFLD is complicated due to the multifactorial nature of its pathogenesis. While the existence of a metabolic disorder plays a role, other factors in a metabolically healthy individual may contribute to NAFLD development. Several genetic mutations have been identified as playing a role in NAFLD development and progression to NASH, including mutations in the Patatin-Like Phospholipase Domain Containing 3 (PNPLA3) and Membrane-Bound O-Acyltransferase Domain Containing 7 (MBOAT7) genes [3]. Furthermore, a growing body of evidence supports an epigenome contribution to NAFLD development and susceptibility [4].

Scientists have come to recognize the crucial role of non-coding RNA (ncRNA) in cellular functions. NcRNA is a class of RNA that does not code for proteins and comprises most of the human genome. Since its discovery, ncRNAs have been used in therapeutic interventions and as diagnostic biomarkers for multifactorial diseases, including liver disorders [5]. Several types of ncRNAs have been shown to contribute to chronic liver diseases, including micro-RNA (miRNA), long non-coding RNAs (lncRNAs), P-element-induced wimpy testis (PIWI)-interacting RNAs (piRNAs), and circular RNAs (circRNAs). The mechanism of action varies for different ncRNAs when regulating gene expression. LncRNA acts at the genomic level and participates in chromatin remolding and histone modification, which alter gene expression [6]. MiRNA plays an essential role in post-translational modification by interacting with the 3′-untranslated region (3′UTR) of target mRNAs and inducing the degradation or hindering the translation of the target mRNA into a functional protein [7]. Upon its discovery, circRNA was known to mainly function as an miRNA sponge leading to miRNA suppression. CircRNAs were also found later to interact with proteins involved in alternative splicing and gene transcription fields [8]. PiRNAs were initially identified as regulators of transposon silencing in *Drosophila* ovarian germline cells [8]. PiRNAs were shown later to regulate mammalian gene expression by interacting with the PIWI protein family and forming a piRNA-induced silencing complex (piRISC). PiRISC plays a vital role in protein expression, genome rearrangement, and reproductive stem-cell maintenance [9].

With the global rise of NAFLD prevalence, it is anticipated that NASH will also become increasingly burdensome. Consequently, increased morbidity and mortality associated with liver disease are also expected. Understanding molecular mechanisms underlying the disease will provide a means to develop effective therapeutic interventions and may provide insight into how these conditions can be prevented from developing in the future. In this review, we discuss the pathogenesis of NAFLD, the involvement of ncRNA in NAFLD/NASH disease, and how different ncRNAs can participate in the modulation of the liver micro-environment concerning high-fat influx into hepatocytes. This review provides insight into how ncRNA acts as a regulatory molecule in metabolic pathways associated with NAFLD initiation and transition to NASH (Figure 1).

## 2. Liver Cellular Function and Development of NAFLD/NASH

The liver is composed of several cell types; every cell population has a distinct function in regulating liver function and homeostasis. Most liver cells are hepatocytes (HCs), comprising approximately 80–85% of the liver mass. The rest of the liver mass is made up of Kupffer cells (KCs), hepatic stellate cells (HSCs), liver sinusoidal cells (LSECs), and cholangiocytes (biliary cells) [10]. Studies have shown that all liver cells can be involved in the development of and progression to NAFLD with each cell type having a distinct role in the pathology of disease progression.

The primary functional unit of the liver is the hepatocyte in which most of the cellular functions take place, including the metabolism of macromolecules (carbohydrate, lipid, and protein), the execution of detoxification pathways, and the activation of immune cells. The pathological role of the hepatocytes in developing NAFLD/NASH is well documented, and studies agree that hepatocytes can act as an inflammatory activator in the liver [11]. With most metabolic activity occurring in the hepatocytes, these cells can be exposed to various stressors in the regular physiological state and are vulnerable to injury. The ingestion of toxins, drugs, alcohol, and dietary overload can all lead to hepatocyte injury, which results in the activation of several inflammatory cascades. Injured hepatocytes are more susceptible to cell death and thus trigger the release of reactive oxygen species (ROS), inflammatory cytokines, and repair signals. This also leads to hepatocyte organelle dysfunction, including mitochondrial damage, lysosomal disruption, and endoplasmic reticulum stress. Ultimately, this increases the intercellular communications between the hepatocytes and the other cells within the liver, especially the KCs [12,13,14]. Kupffer cells are the liver-resident macrophages. KCs release several chemokines and cytokines upon activation to enhance their polarization in physiological and pathological conditions. Persistent inflammation of the liver tissue increases the activation of the pathological type of KCs, eventually leading to hepatocellular damage [15]. Injured sinusoidal cells caused by liver steatosis can also activate KCs. Injured LSECs are inflammatory mediators that contribute to the recruitment of other inflammatory cells, leading to liver injury [16]. Stellate cells, conversely, are dynamic and can be found in inactive or active states. Quiescent stellate cells in the healthy liver act as a storage site, mainly for fat and vitamin A as a retinyl ester. In contrast, the active cells act as the primary fibrogenic cells during liver injury [17].

Intercellular communications between liver cells are complex and dependent on the microenvironmental state. Cell response to metabolic or immune stimuli will change over time depending on the type of signal released, the gene expression, and the duration of the inflammatory event [18]. Several types of liver cells are implicated in the complex process involved in the development of NAFLD and subsequent progression to NASH.

### 2.1. Pathogenesis and Clinical Aspects of NAFLD/NASH

NAFLD is clinically characterized by simple steatosis (SS). The progression of the disease into non-alcoholic steatohepatitis (NASH) may lead to the development of other clinical pathologies, including liver fibrosis, cirrhosis, or hepatocellular carcinoma (HCC) (Figure 2). NAFLD results from multiple parallel factors including genetic, environmental, and metabolic components that act synergistically in disease development. Initially, the liver becomes infiltrated with excess fat due to lipid metabolism imbalance, often due to high caloric intake, a Western diet, and lifestyle factors. Patients with NASH/NAFLD develop insulin resistance as the disease progresses. Insulin resistance in adipose tissue leads to enhanced lipolysis in adipocytes. As a result, adipose tissue releases high amounts of free fatty acids. The dysregulation of lipid balance leads to triglyceride accumulation in hepatocytes causing the characteristic “fatty liver.” Storing triglycerides in and of itself does not harm the liver, but it does indicate that the hepatocytes are exposed to fatty acids that could potentially cause harm [19,20]. In NAFLD patients, increased fatty acid uptake and de novo lipogenesis (DNL) compromise the body’s ability to oxidize or export fatty acids, leading to excessive triglyceride accumulation in the liver [21].

Triglyceride accumulation in hepatocytes is a clinical characteristic of NAFLD and NASH. NAFLD progression to steatohepatitis occurs in hepatocytes due to an elevated level of DNL, decreased oxidation, and a decreased very low-density lipoprotein (VLDL) secretion which may result in lipotoxicity [19]. Hepatocyte death is one of the most significant pathological events in liver disease, mainly occurring as a homeostatic reaction to internal or external disruptions to eliminate damaged cells [22]. When adaptive systems that protect hepatocytes from fatty acid-mediated lipotoxicity become overburdened, repair responses may be triggered, which may include the activation of hepatic stellate cells, Kupffer cells, and myofibroblasts. Activated HSCs play a key role in the development of liver fibrosis. Upon activation, HSCs produce remodeling (factors for tissue repair such as matrix metalloproteinases (MMPs) and their tissue inhibitors (TIMPs). A substantial amount of extracellular matrix (ECM) will continue to be produced by prolonged HSC activation leading to the attraction of other immune cells, especially macrophages. Interplay between the activated HSCs and the macrophages has long been suspected to have a crucial role in the development of fibrotic liver. Both cells can produce proinflammatory and profibrogenic cytokines that support the inflammatory response within the liver. The activated macrophages have been shown to support the survival of HSCs and the continuous activation of the fibro-genic pathway [23,24,25].

Myofibroblasts create an excessive amount of matrix and other substances that promote the growth of hepatic progenitor cells. Progenitor cells then produce chemokines to attract various inflammatory cells to the liver; they also differentiate to replace dead hepatocytes. The repair response often corresponds to the degree of hepatocyte mortality, resulting in varying hepatic architectural distortions with fibrosis, invading immune cells, and regenerating epithelial nodules. Liver cirrhosis develops in people with NAFLD when the repair is intense and prolonged but fails to reconstitute healthy hepatic epithelia [19,26]. The interaction between liver cells is illustrated in Figure 3.

In simple steatosis, DNL, due to high free fatty acid (FFA) uptake by the liver, is controlled by insulin signaling via activation of sterol regulatory element-binding protein 1c (SREBP-1c) and protein kinase B (AKT) signaling [27,28,29].

The expression of SREBP-1c depends on the activity of the liver X receptor (LXR) and peroxisome proliferator-activated receptors (PPARs), both of which respond to dietary cholesterol. It has been reported that the activity of SREBPs is contrariwise to NAFLD/NASH severity. The study group showed that SREBP expression is significantly increased in a NASH-HCC mouse model in the simple steatosis but not in the advanced stage. This finding suggests the importance of SREBPs in lipid homeostasis in the liver [30]. LXR activation was achieved by cholesterol supplementation leading to active SREBPs suggesting that LXR activity depends on lipid influx which in turn activates SREBPs [31]. LXR-SREBP activity was also examined in a mouse model exhibiting low expression of SREBPs. The study showed that LXR activity was lower in SREBP knock down mice in comparison to the wild type [30].

Several enzymes involved in FFA metabolism will be activated upstream to SREBP, mainly consisting of Acetyl-CoA carboxylase (ACC) and fatty acid synthase (FAS). The synthesis of triglycerides (TAGs) will be enhanced through the activation of glycerol-3-phosphate acyltransferase. As the TAGs accumulate in the hepatocytes, the amount of stress on the endoplasmic reticulum and the mitochondria will prompt the innate immune response, activating Toll-like receptor 4 (TLR4). TLR4 will respond to the stress signals and induce the release of several cytokines from Kupffer cells, thus mediating cell injury and cell death.

The dysregulated immune response is a hallmark of chronic inflammatory liver diseases including NAFLD. Prolonged activation of the innate immune response is a result of improper lipid metabolism in the liver [32]. It induces hepatic insulin resistance (IR). This is primarily caused by activated TLR4, which inhibits AKT phosphorylation and increases the production of proinflammatory cytokines, especially tumor necrosis factor-alpha (TNF-α) and interleukin-6 (IL-6). Both cytokines will activate nuclear factor-κB (NF-κB) and enhance IR. Persistent insulin resistance within the liver and excessive FFA and DNL activation will trigger fibrogenic and apoptotic pathways. This leads to transforming growth factor β (TGF-β) production, collagen deposition by activated HSCs, and the eventual progression of NASH to fibrosis. The state of a fibrotic liver is mostly irreversible and will become cirrhotic in the long term. Cirrhosis represents the end stage of chronic inflammation and subsequent decompensation has a greater chance of resulting in hepatocellular carcinoma (HCC) [33,34,35].

### 2.2. Current Treatment Options for NAFLD/NASH

NASH-related HCC is a severe complication, and efforts have been widely implemented to reduce the burden of related morbidity and mortality. Patients with suspected NAFLD/NASH are subjected to risk assessment using non-invasive fibrosis markers, including the Fibrosis-4 score or the FIB-4 test, which has shown high negative predictive value to exclude patients with advanced fibrosis. When the FIB-4 test results in a score less than 1.3, primary care can manage a patient with counseling about lifestyle modification and clinical assessments every one-to-two years. Patients who have type two diabetes, who are prediabetic, or who have two or more metabolic risk factors may require more frequent assessment and evaluation given their higher risk of disease progression [36]. Patients with a FIB-4 score greater than or equal to 1.3 but less than 2.67 are usually referred to a hepatologist for further assessment to rule out advanced fibrosis or cirrhosis. Further non-invasive testing includes the enhanced liver fibrosis test (ELF), MR Elastography, Vibration controlled transient elastography (VCTE), and Fibroscan or corrected T1 (cT1). A liver biopsy will be indicated if noninvasive test results are indeterminate or if there is a need to rule out other underlying causes of liver disease [37].

Currently, there is no approved treatment that has been identified for NAFLD/ NASH, and current treatment options involve medications that are effective only at the progression of the disease. Weight loss achieved through adjusting lifestyle and diet is associated with improved steatosis and fibrosis. Drugs used for type 2 diabetes such as Vitamin E and pioglitazone, and Glucagon-like peptide-1 agonists (GLP-1) were also shown to be effective in controlling NAFLD/NASH symptoms [38].

Some compounds, such as vitamin E, have been shown to potentially impact NAFLD/NASH [39]. Health practitioners are interested in dietary supplementation in the management of several diseases including NAFLD/NASH [40]. Currently, there is an ongoing clinical trial (ICAN, NCT04781933). It uses a combination of dietary supplements including vitamin E, probiotics, curcumin, and other plant extracts in an effort to prevent fibrosis in NASH patients.

Pioglitazone, a drug taken for type two diabetes mellitus (T2DM) management, was associated with improvement in symptoms of steatosis/NASH in patients with and without diabetes [41]. GLP-1, including liraglutide and simaglutide are approved for use in obese patients with T2DMand in NASH patients without cirrhosis with a definite improvement in steatosis but no proven benefit for fibrosis [42]. Recently, GIP/GLP-1 receptor agonist, tirzepetide, has been approved for type 2 diabetes patients and for obese patients with NAFLD who showed improvement in insulin sensitivity and steatosis [43]. Several clinical trials have been conducted to assess the safety and tolerability of tropifexor, a farnesoid X receptor (FXR) agonist, alone or in combination with other drugs. Despite the termination of several trials due to adverse events, the use of tropifexor has shown promising improvement in NASH patients with regard to the hepatic fat fraction in NASH patients [44,45].

## 3. NcRNA and Intercellular Communication in Liver Cell Populations

With the advancement in RNA-sequencing technologies (RNA-seq), several studies have shifted focus to identify functional RNA transcripts that can reveal molecular mechanism dysfunction in liver diseases, including NAFLD. These functional or ncRNAs are classified based on length, structure, function, and genomic location. Several classes of ncRNA have been found to modulate gene expression within the liver and to participate in protein regulation. Among these ncRNAs are miRNA, piRNA, ciRNA, and lncRNA.

Intercommunication between liver cells is a complex process in both physiological and pathological states. It could be assumed that the proinflammatory environment within the liver caused primarily by HSC signals will activate the resident liver macrophages (i.e., KCs) and possibly other cell types involved in NAFLD/NASH pathophysiology. The ncRNA molecules play a crucial role in regulating the inflammatory responses of liver cells in healthy and compromised liver tissue.

### 3.1. MiRNA

MiRNA has been shown to play a key role in the pathogenesis of NAFLD. Patients with NAFLD were shown to have distinct miRNA signatures in liver tissue and peripheral blood. Hepatocytes of NAFLD patients become insulin-resistant cells in response to high fat deposition and those HCs are responsible for initiating either protective or pathogenic signals, including exosome secretion. It has been found that extracellular vesicles (EVs) secreted by HCs in response to internal stimuli have a unique miRNA profile [46]. In the acute phase of fatty liver disease, Ji et al. found that EVs derived from HCs express miR-3075, which inhibits fatty acid 2-hydroxylase (FA2H), an enzyme responsible for the hydroxylation of free fatty acids, and enhances the hepatocyte insulin sensitivity in vivo. In contrast, chronic accumulation of fats in the hepatocytes resulted in the loss of miR-3075 expression and promoted insulin resistance. This finding suggests an anti-inflammatory role for miR-3075 in simple steatosis [46]. Another study found that HC-derived EVs were enriched by miR-192-5p [47]. MiR-192-5p increased KC polarization and enhanced inflammation by targeting nitric oxide synthase (NOS) expression. NOS overexpression increases IL-6 secretion by resident KCs and induces inflammation [48]. Pourhoseini et al. reported on the vital role of miR-21 in sinusoidal injury related to NAFLD/NASH in vivo. The increased expression of miR-21 led to increased NOS phosphorylation in liver sinusoidal cells and enhanced LSECs dysfunction [49]. Interestingly, the study group elucidated the importance of the crosstalk between liver cells. Stellate cells have been found to initiate the inflammatory response caused by miR-21 that results in the activation of KCs and LSECs injury. Furthermore, lipotoxic HC-derived EVs showed a high level of miR-9-5p and miR-1297 in comparison to normal cells. These miRNAs were experimentally shown to enhance KC polymerization and HSC activation by targeting transglutaminase 2 (TGM2) and the PTEN signaling pathway, respectively [50,51]. Recently, it has been widely accepted that miRNA-122 is a hepato-specific miRNA, and its role in the progression of steatosis to steatohepatitis is being thoroughly investigated [52,53]. Moreover, miR-122 was found to be highly expressed in hepatocytes, and it targets lipogenic genes to decrease hepatic fatty-acid oxidation [54,55,56]. The inhibition of miR-122 levels enhances fatty acid oxidation, decreases lipid deposition in the hepatocyte, and reduces inflammation. Interestingly, it was shown that increased miR-122 levels led to the activation of KCs. In contrast, the downregulation of miR-122 reduces KC activation and enhances immune homeostasis within the liver [57,58].

### 3.2. LncRNA

LncRNAs are another class of ncRNA found to be associated with NAFLD development. Of note, the steroid receptor RNA activator (SRA) is lncRNA, which was primarily found to enhance steroid nuclear receptor-dependent gene expression [58,59]. In 2016, SRA was reported to be implicated in lowering HC FFA beta oxidation via modulation of forkhead box protein O1 (Fox01) expression in SRA-knockout mice [60]. Shortly after, another study reported a high expression level of brown fat-enriched lncRNA 1(Blnc1) in HCs of obesity-induced animal models. Remarkably, the knockout of Blnc1 in vivo resulted in lowering hepatic lipogenesis, reducing inflammation, and decreasing cell apoptosis [61]. Another lncRNA known as metastasis-associated lung adenocarcinoma transcript 1 (MALAT1) was also found to regulate HC gene expression. The nuclear sterol regulatory element-binding protein (SREBP)-1c belongs to the SREBP family that regulates lipid metabolism by controlling enzymes involved in lipolysis/lipogenesis. The expression level of SREBP-1c in HCs acquired from individuals with NAFLD was significantly higher compared to the normal cell population [62]. An in vivo study by Yan et al. found that MALAT1 is upregulated in HCs of type 2 diabetic mice. Using RNA pull-down assay, the same study found that MALAT1 interacts with SREBP-1c and increases hepatic steatosis. This will in turn lead to hepatocyte dysfunction and liver inflammation. It has also been found that MALAT1 stimulates the fibrogenic pathway via activation of hepatic stellate cells by directly activating CXCL5, a CXC subfamily of chemokines [63].

Chronic hepatic inflammation is a hallmark of liver steatosis that can lead to the activation of liver resident macrophages KCs. The correlation between lncRNA expression and KC activation has been reported by Zhang et al. [64]. Zhang and his group studied a liver-enriched lncRNA known as Lfar1. The investigators reported that Lfar1 is responsible for activating liver macrophages in response to HC death and HSC activation. Hepatic macrophage activation can also be achieved indirectly via Cholangiocyte-derived exosomal lncRNA known as H19 [65]. This lncRNA can be expressed in KCs, leading to cell activation and inflammatory cytokine secretion. Activated KCs will induce hepatic inflammation and tissue damage by increasing HC programmed cell death and stellate cell trans-differentiation [66]. These studies suggest the critical role of lncRNA in the induction of liver injury caused by the HC dysfunction, the stellate cell activation, and the KC polymerization that is seen in NAFLD/NASH.

Nuclear Enriched Abundant Transcript (NEAT1) is highly prevalent in the liver and serves as a risk factor impacting NAFLD. It has the potential to expedite the progression of NAFLD by promoting the accumulation of lipids in the liver [67]. In regard to hepatic lipid regulation, a study revealed that the lncRNA NEAT1 augmented the expression of adipose triglyceride lipase by competitively binding to miR124-3p. This competitive binding subsequently elevated the levels of diacylglycerol and free fatty acids. The interaction between NEAT1 and miR-124-3p disrupted the process of lipolysis and increased fatty acid oxidation through the PPARa signaling pathway in liver cells, as reported by Liu et al. in 2018 [68]. Another study conducted by Bu. et al. has demonstrated that NEAT1 is linked to the regulation of the mTOR/S6K1 signaling pathway, which plays a crucial role in lipid biosynthesis and insulin signaling transduction. The depletion of NEAT1 has been shown to impact the mRNA and protein expression of f acetyl-CoA carboxylase enzymes and fatty acid synthase through the modulation of the mTOR/S6K1 signaling [69].

### 3.3. CircRNA

CircRNA is a non-linear RNA that acts as a miRNA sponge and to regulate gene expression. Functional studies have been conducted to elucidate the role of circRNA in many diseases, including NAFLD. CircRNA involvement in NAFLD/NASH is still being investigated. It was reported that about 93 circRNAs were differentially expressed in the liver tissue of a NAFLD mouse model, suggesting a role in liver steatosis [70]. In addition to its expression, the localization of circRNA may have a vital role in NAFLD progression and transition to NASH [71]. Mitochondrial circRNA, known as steatohepatitis-associated circRNA ATP5B regulator (SCAR), contributed to NAFLD/NASH in vivo [71]. CircRNA SCAR may participate in the activation of liver fibroblasts and the enhancement of NASH-related fibrosis. The cytoplasmic location of other circRNA, namely circRNA_0046367, was found to be lost due to hepatocyte damage resulting from mitochondrial dysfunction [72].

Free fatty acid treated primary human hepatocytes had a low expression level of circRNA_0001805 and were more susceptible to inflammation [73]. Transfection of cells by circRNA_0001805 effectively reduces hepatic inflammation, suggesting a preventive role for this circRNA in steatosis-derived inflammation. Recently, miR-122 expression was reported to be regulated by a circRNA known as circPI4KB [74]. The study group reported significant findings showing that circPI4KB translocated miR-122 from the hepatocyte to the extracellular space, which results in low expression of miR-122 in the hepatocyte and subsequently induces fat deposition. These findings contradicted the widely accepted role of miR-122 in lipid metabolism within the hepatocyte.

Qiang and his team described a critical role for circRNA_29981 in HSC activation via interaction with miR-181b [75,76,77]. Two more circRNAs, hsa_circ_0071410 and circBNC2, were found to activate HSCs and HCs, and they may play a role in the liver fibrosis seen in NASH patients [76,77,78]. In contrast, circFBXW4 has the opposite effect and has been shown to act as an HSC suppressor, which inhibits the fibrogenic pathway by interaction with miR-18b [14].

It has been reported by Chen et al. that the expression of circDIDO1 decreased in HSCs subjected to radiation [76]. Another study utilized this finding to investigate further the role of circDIDO1 in fibrosis-related processes. Their findings demonstrated that overexpressing circDIDO1 suppressed the growth and fibrosis of HSCs. This effect was achieved by reducing pro-fibrotic markers such as a-SMA and collagen I, inducing apoptosis, and causing cell cycle arrest in HSCs. The study also confirmed that the circDIDO1/miR-143-3p axis played a crucial role in activating the PTEN/AKT pathway [79].

Another research study suggested that Exo-circCDK13 has the potential to alleviate liver fibrosis by suppressing the PI3K/AKT and NF-κB signaling pathways through the regulation of the miR-17-5p/KAT2B axis. This could directly impact the protein expression of collagen I and fibronectin [80].

### 3.4. PiRNA

PiRNAs were named after their biological function. These molecules are small ncRNAs that interact with Piwi-Argonaute proteins in the germ line [81,82]. PiRNAs function as transposon silencing elements in addition to their role in spermatogenesis [83]. The role of piRNA in metabolic diseases is not well investigated, and to our knowledge, only a limited number of studies have been conducted in relation to NAFLD.

Using microarray technology, differential expression of piRNA was reported using liver tissue from NAFDL mice [84]. This study suggested piRNA may have a regulatory role in NAFLD by interacting with proteins involved in several metabolic pathways. Using in-silico analysis, the study highlighted the association of dysregulated piRNA with cellular components of liver tissue and suggested piRNAs may act as liver cell activators.

In 2018, Tang et al. studied the correlation between piR-823 and HSC activation [85]. Loss and gain of function assays using piR-823 antagomir or piR-823 mimic significantly impacted HSC behavior. While piR-823 overexpression increased hepatic stellate cell proliferation, its inhibition increased the proportion of quiescent cells by targeting TGF-β [85]. These results suggest piRNA is vital in HSC activation and the likelihood of developing a fibrotic liver. Furthermore, it is well known that macrophages, including KCs, are the main cells that can drive, sustain, or resolve inflammation within a given tissue. Recently, it was found that several ncRNAs, including piRNAs are secreted by macrophages and encapsulated in EVs. These ncRNA molecules were thought to modify the immune response within the tissue microenvironment and to enhance or prevent inflammation by activating other cell types [86,87]. Due to limitations in the analysis pipelines used in piRNA annotations, misinterpretation of the data can occur, affecting the overall study conclusion [88].

## 4. NcRNA and the Dysregulated Pathways Related to NAFLD/NASH

The vicious cycle of insulin resistance and lipid accumulation leads to a cascade of biological events as NAFLD progresses to NASH, including oxidative stress, lipotoxicity, inflammation, and eventually fibrosis. ncRNAs, including lncRNAs, circRNAs, and miRNAs, are known to be significant regulators in the pathogenesis of NAFLD as indicated by high-throughput sequencing methods [67]. More studies are needed to fully comprehend how these molecules interact within and between the hepatocytes with regard to pathogenesis. A list of ncRNAs involvement in NAFLD is listed in Table 1.

### 4.1. NcRNA, Lipotoxicity and Oxidative Stress

Under physiological condition, lipid metabolism occurs in a state of equilibrium, where the catabolism and anabolism of lipids are regulated based on the needs of the cell. Three major classes of lipids exist and are utilized by cells to maintain their function. The lipids include glycerolipids, phospholipids and sterols all classified according to the backbone of their structure. Lipid metabolism occurs mainly in the liver, where a constant amount of free fatty acids is influxed into the HCs. The cell’s need dictates the biosynthesis rate of different classes of lipids. However, their existence is also affected by the pre-existence of a metabolic disorder [110]. The interruption of lipolysis and lipogenesis pathways within the liver is a recognized risk factor contributing to chronic liver disease, including NAFLD. This dysregulation of lipid turnover may increase FFA concentration in the hepatocytes, leading to TAG accumulation and hepatic steatosis. Excessive lipids in the liver can lead to lipotoxicity and eventually to the loss of function of organs, primarily resulting in mitochondrial failure and endoplasmic reticulum stress, which leads to metabolic inflammation and, subsequently, NAFLD and NASH pathogenesis [111,112].

Normally, HC lipid metabolism is carried out in two main pathways: beta-oxidation or esterification. Beta-oxidation breaks down FFA into acetyl-CoA units while esterification is TAG formation from FFA when it interacts with glycerol-3-phosphate (G3P) [113]. In lipid metabolism, several ncRNAs have been identified as regulators of lipid hemostasis in the HCs by targeting key molecules in both pathways to prevent or enhance lipotoxicity.

As mentioned, sterol regulatory element binding proteins (SREBP) are a protein family of transcription factors that act as central molecules in lipid metabolism [114]. This protein family controls the expression levels of several enzymes in the endogenous synthesis of phospholipids, cholesterol, and triglycerides [115]. It has been reported that miR-124 expression correlates with SREBP-1c isoform in vivo, suggesting that high-fat hepatic content stimulates the SREBP-1c expression along with miR-124 [116]. The inhibition of miR-124 resulted in low expression of SREBP-1c and other lipogenic genes, including fatty acid synthase (FAS) and Acetyl-CoA synthetase (AceCS). Three other miRNA molecules: miR-96, miR-182, and miR-183 were also found to regulate the same protein family in vivo and in vitro [89]. This group of miRNAs is transcribed from the same miRNA locus, and SREBP controls their expression levels. SREBP was shown to directly bind the promoter region of this miRNA cluster and transcriptionally activate their expression. These data suggest the predominant role of miRNA/SREBP-dependent lipid homeostasis within HCs. Furthermore, lncHR1 was shown to regulate SREBP [93].

Liver X receptors (LXRs) are nuclear receptors that serve as cholesterol sensors with an active role in response to high cholesterol efflux into the cells. In addition to their main function, LXRs are also thought to increase TGA and liver steatosis [117]. Sun and his group reported that miR-26 expression is LXR dependent, suggesting that activated LXR signaling induced by lipotoxicity suppresses the protective role of miR-26 [95].

Peroxisome-proliferator-activated alpha (PPARα) is another nuclear receptor highly expressed in the liver. These anti-steatosis proteins protect the liver from lipotoxic insults [118]. Normally, FA transportation to the liver will induce mitochondrial β-oxidation to produce energy, but when the mitochondria are overwhelmed, the proliferation of peroxisomes is activated instead, leading to an increase in overall ROS amount [118]. An elevated ROS production leads to an imbalance between reactive oxygen species and protective oxidant production, resulting in oxidative stress and proinflammatory events associated with NASH [111,119]. Investigators in a study published in 2018 found that PPARα signaling dysfunction resulting from lipid overaccumulation might be corrected by two ncRNAs: miR-34a and circRNA_0046366. The study found that circRNA_0046366 expression is lost in hepatocellular steatosis while miR-34a is expressed. High expression of miR-34a resulted in PPARα dysfunction and accelerated hepatic lipid accumulation. Using a luciferase assay, circRNA_0046366 was found to interact with miR-34a, and restoration of its expression resulted in miR-34a repression and PPARα normalization. It appears that miR-34a and circRNA_0046366 may be involved in the normal function of PPARα, assisting in reducing hepatic fat content and stimulating the antioxidant defense mechanism in the liver [96].

Another signaling pathway that also participates in lipid metabolism involves phosphatidylinositol-3 kinases (PI3K)-Protein kinase B (Akt)-mammalian target of rapamycin (mTOR). Under lipotoxic conditions, the PI3K/AKT/mTOR pathway is induced leading to mitochondrial dysfunction and increased oxidative stress [90,120,121]. The activated state of this pathway seems to positively correlate with the expression of lncARSR and to determine the overall cholesterol biosynthesis in the hepatocytes. The knockdown of lncARSR in the mice model reverses the harmful effects that activated the PI3K/AKT/mTOR pathway, lowering the rate of hepatic Ch via targeting HMG-CoA reductase [99]. Crosstalk between PI3K/AKT/mTOR, SREBPs, and PPARα is well established by several studies, highlighting roles in metabolic regulation [122,123]. Recently, miRNA33a and b* have emerged as pivotal regulators of metabolic programs such as cholesterol and fatty acid homeostasis. Extensive research has focused on these two miRNAs leading to the identification of novel targets. It is now established that the effects of miR-33a and miR-33b target genes governing cholesterol metabolism, fatty acids β-oxidation, and insulin signaling [124]. MiR-33a/b plays a specific role in regulating cholesterol transport pathways, facilitating the mobilization of cholesterol from intracellular stores to HDL lipoproteins. This crucial miRNA actively engages in the metabolism of fatty acids and cholesterol through collaboration with transcription factors SREBP-1 and -2 (sterol-regulatory element-binding protein-1 and -2), respectively. Moreover, it plays a role in regulating the expression of genes associated with lipid and cholesterol synthesis [125].

NcRNA coordinates the cell response to toxic signals, including lipids. Several mechanisms have been suggested, but more functional studies are required to elucidate the precise mechanistic role of ncRNA in response to lipotoxicity.

### 4.2. NcRNA, Inflammation, and Fibrosis

Inflammation is a host immune system defense response to danger signals. The duration of the event determines the classification of the activated inflammatory response. An acute inflammatory response is a quick immunological reaction that causes blood vessel dilation and an increase in the population of inflammatory cells, proteins, and fluid at the injury site. A chronic inflammatory response might continue for months or years due to sustained exposure to injury. In the case of NAFLD/NASH, innate immune activation plays a critical role in generating and increasing hepatic inflammation [111]. Inflammasome activation, the release of hepatocyte-derived signals, altered innate immune signaling, and altered macrophage, T cell, platelet, and neutrophil responses have all been found to play a vital role in acute and chronic liver illnesses. Furthermore, NAFLD and NASH are considered cytokine-driven diseases because multiple proinflammatory cytokines (IL-1, IL-1, TNF-alpha, and IL-6) play key roles in inflammation, steatosis, fibrosis, and the development of hepatocellular carcinoma. The inflammation must be cleared or controlled to prevent the pathological consequences of the inflammatory response seen in NAFLD patients. Several methods have been tried to prevent NAFLD-associated inflammation, such as using a vitamin E supplement. NcRNA, in particular, was shown to avoid or enhance existing inflammation in vitro and in vivo. This study could contribute to the discovery of potential biomarkers and prognostic factors or the advancement of therapies [111].

Nuclear factor kappa B (NF-κB) is among the well-studied signaling pathways that act as a dominant mediator for inflammation. NF-kB can be activated through several stimuli. Activation of NF-kB by the tumor necrosis factor receptor (TNF-R) and the NOD-like receptor family pyrin domain-containing 3 (NLRP3) is well and reported in NAFLD/NASH models. Studies have shown that activated NF-κB can induce or inhibit miRNA expression. Shen et al. showed a correlation between increased miR-155 expression and active NF-κB levels in NAFLD-rat model, suggesting that NF-κB could induce miR-155 expression and enhance inflammation [100]. Furthermore, it has been reported that miR-155 hampers the expression of several transcripts that act as a TNF-α suppressor [101]. Moreover, miR-125 was also found to promote NAFLD inflammation via activation of NF-κB. MiR-125 was overexpressed in liver tissue and inhibits the translation of proteins involved in regulating the TNF apoptotic pathway [104]. Altogether, miR-125, miR-155, NF-κB, and TNF play a central role in multiple stages of NAFLD development and in the progression to NASH. Activity remodeling of NF-κB by miRNA in the liver is a potential strategy for preventing or delaying the onset of NASH in an individual with NAFLD (Figure 4).

Research on mice with steatohepatitis induced by a methionine-choline-deficient (MCD) diet has yielded findings that may indicate a self-protective mechanism against steatohepatitis. This mechanism involves the inhibition of NF-κB. In cases of steatohepatitis, an activated NF-κB pathway drives the transcriptional expression of lncRNA known as Platr4 (pluripotency-associated transcript 4), which subsequently hinders NF-κB activity. Moreover, it deactivates the NlRP3 inflammasome by preventing the binding of NF-κB to κB sites in the promoters of target genes, including NlRP3. The same study found that the alleviation of steatohepatitis in mice overexpressing Platr4 was associated with decreased hepatic levels of NlRP3 mRNA and reduced production of IL-1β and IL-18 proteins. These results suggest that the inactivation of the NlRP3 inflammasome contributes to the preventive effects of Platr4 on NASH [102]. Another lncRNA that has been associated with the pathogenesis and progression of NAFLD to NASH is lncTNF. lncTNF is an intergenic lncRNA that plays a role in liver inflammation, demonstrating a robust response when subjected to TNF-α stimulation [103]. Several studies have shown an upregulation of lncTNF in NAFLD, and the main function associated with it is to promote inflammation via the NF-κB pathway [103]. However, lncTNF is considered a novel ncRNA and studies regarding the involvement of lncTNF are limited. Although the role of PiwiRNA in the chronic inflammatory response was also studied in varying inflammatory diseases, the direct correlation between NAFLD-related inflammation and piRNA has not been thoroughly investigated. However, PiwiL2 and PiwiL4 expressions were reported to correlate with TNFα and IL1β exposure. Both TNFα and IL1β are proinflammatory mediators that are deregulated in NAFLD.

The involvement of ncRNA in the transition of steatohepatitis to fibrosis also plays a role in this process. Several biological pathways and cells are involved in the development of fibrosis. As mentioned earlier, active stellate cells can primarily become fibrogenic cells upon liver injury [17]. Furthermore, studies have shown that activated HSCs exacerbate liver fibrosis in NASH through the up-regulation of TGF-β1, [126,127]. Proteins such as the ones in the SMAD protein family are well-established NAFLD mediators downstream of the TGF signaling pathway [126]. Numerous research investigations have highlighted the involvement of both miR-21 and miR-29b in liver fibrosis by interaction with TGF-β1/SMAD and collagen type I alpha 1 chain, respectively [94].

CircRNA known as circFBXW4 was studied in the activated hepatic stellate cells derived from fibrogenic liver tissue. Remarkably, Chen and his group demonstrated a loss of circFBXW4 expression in HSCs as well as in patients with hepatic fibrosis. Upon the overexpression of circFBXW4 in vivo, the HSCs were deactivated which resulted in fibrogenic factor inhibition. This was followed by a decrease in the circulating levels of proinflammatory cytokines TNF-α and IL-1β, suggesting the anti-fibrotic role of circFBXW4 [14].

H19 is a lncRNA that exhibits paternal imprinting and maternal expression. It was among the earliest identified lncRNAs. More recently, more studies have highlighted the extensive involvement of H19 in the pathological mechanisms associated with fibrosis [105]. H19 levels were found to be downregulated in rats with induced fibrotic liver tissues, particularly in TGF-β1-activated HSCs [105]. Moreover, it has been hypothesized that the lncRNA Alu-mediated p21 transcriptional regulator (APTR) is implicated in hepatic fibrogenesis. The reduction in APTR prevents collagen formation by blocking the TGF-β-dependent induction of α-SMA in vivo [128]. Several studies have deduced that the levels of APTR are elevated in both fibrogenic animal models and in patients [128,129].

Zhang et al. have observed while examining lncRNAs from the livers of fibrotic mice that the level of a specific lncRNA, known as lnc-LFAR1, is reduced in the entire liver but notably increased within HSCs during the development of fibrosis. The action of TGFβ triggers this increase in lnc-LFAR1 expression and plays a role in activating HSCs while also contributing to the apoptosis of HCs induced by TGFβ. Their research revealed that lnc-LFAR1 facilitated the interaction between SMAD2/3 proteins and TGFβR1, subsequently leading to the phosphorylation of SMAD2/3 in the cytoplasm. Additionally, they confirmed findings of the interaction between lnc-LFAR1 and the transcription factors SMAD2/3 through RNA immunoprecipitation (RIP) assays. Their findings indicate that the suppression of lnc-LFAR1 significantly disrupts the fibrotic TGFβ/SMAD pathway in both HSCs and HCs, reducing liver fibrosis [130]. These data may have prompted some studies aiming to identify drugs and potentially randomized clinical trials will soon follow. In primary HSCs isolated from CCl4-induced fibrotic liver at various time points, Yu et al. observed an abnormal upregulation of NEAT1. Simultaneously, NEAT1 exhibited a gradual increase during the differentiation of primary HSCs. Silencing NEAT1 via Ad-shNEAT1 delivery suppressed CCl4-induced collagen deposition in mouse liver and hindered the activation and proliferation of primary HSCs in vitro [131].

The literature needs to revisit piRNA species and their association with hepatic fibrosis mechanisms and pathways. Tang et al. investigated the effect of piR-823 expression in mice HSCs revealing that an increase in piR-823 expression led to the production of TGF-b1. Crosstalk between piR-823, TGF-b1, and activated HSCs contributed to the accumulation of extra-cellular matrix (ECM), which is common in liver fibrosis. It is important to note that the development of liver fibrosis is a complex process with multiple contributing factors, and the interaction between these molecules represents just one aspect of this process [85].

## 5. The Role of ncRNA in NAFLD-Associated Metabolic Dysfunction

Recently, it was suggested to rename NAFLD to metabolic-associated non-alcoholic fatty liver disease (MAFLD) [132]. The renaming would reflect the pathological factors that contribute to the development of fatty liver and would provide a comprehensive understanding of the disease progression and management. The new proposed name also highlights the role of metabolic drivers in developing liver steatosis. The established criteria and recommendations for NAFLD screening generally apply to individuals with a pre-excitant metabolic disease, in particular for diabetic or obese patients and people with hyperlipidemia. These metabolic syndromes are considered a risk factor for NAFLD, even in asymptomatic patients. The presence of one or more metabolic risk factors will lead to overall systematic disturbance, insulin resistance, and an increase in lipolysis. Thus, an increase in FFA being sent to the liver will promote hepatic-triglyceride synthesis, reduce FFA oxidation, and enhance fat deposition within the liver [133]. It was estimated that obesity and hyperlipidemia accounted for most metabolic comorbidities associated with NAFLD, with a prevalence of 50% and 70%, respectively. Diabetes accounted for more than 20% of the metabolic comorbidities, while other metabolic induced-NAFLD accounted for approximately 40% of the global prevalence [134].

Obesity is linked to impaired blood glucose tolerance as well as to hyperinsulinemia, which increases the risk of developing diabetes and hyperlipidemia [135]. Obese patients might be present with or without metabolic syndrome, but this will not rule out fatty liver [136]. This complex crosstalk between metabolic syndromes is not fully understood, but it has been proposed that impaired insulin signaling seen in patients with a preexistent metabolic condition is the key player in NAFLD progression. Yet, the dysregulation in insulin secretion or function is linked to multiple metabolic imbalances in the liver, the pancreas, the muscles, and the adipose tissue [137].

A large body of evidence indicates that the role of ncRNA in metabolic modulation involving insulin signaling (Figure 4) has been shown to prevent or contribute to metabolic abnormalities within hepatic cells. Using microarray analysis, Trajkovski et al. identified miR-103 and miR-107 upregulation correlating with insulin sensitivity in liver tissue of obese mice [98]. Interestingly, loss of function analysis of miR-103/miR-107 lowered blood glucose level, enhanced adipose and liver insulin sensitivity, and decreased hepatic glucose production by targeting caveolin-1 (Cav1) mRNA, an essential regulator in glucose and lipid homeostasis.

A study conducted in 2011 showed a significant association between miR-143 and dysregulation of glucose metabolism in a transgenic mouse model through insulin resistance induction [108]. Molecular analysis of the mode of action of miR-143 identified an under-reactive AKT signaling within the liver tissue of obese mice, suggesting that miRNA may impair AKT phosphorylation and reduce insulin sensitivity. Furthermore, miR-143-145 deficient mice showed improved insulin sensitivity and active AKT signaling. In terms of insulin-resistant development, a member of the oxysterol binding protein family (OSBP) known as ORP8, seems to be a direct target for miR-143. OSBP, among other functions, regulates cholesterol efflux and activates AKT signaling. In this study, miR-143-145 deficient mice showed an almost 2-fold increase of hepatic ORP8 compared to control mice, which increased or accelerated hepatic insulin action. Other miRNAs were also found to be deregulated in obese animal models as well as in human hepatic tissue and were found to regulate both glucose and lipid homeostasis which included miR-26a [106], miR-206 [91], and miR-451 [97].

lncRNAs, on the other hand, were also linked to impaired metabolic signaling related to obesity. Yang and his group used integrative transcriptomic approaches to identify lncRNA-mRNA networks crucial for metabolic regulation. The group identified lncRNA Gm16551 to be enriched in the liver, and its expression was lost in obese mice. To determine the nutrient-sensitive lncRNA, nutrient and fasting-induced hepatocytes were used. The expression of three lncRNA including AK085787, uc009kuu.1, and uc008txr.1 seemed to be nutrient-dependent [92] and tended to be induced by insulin. The precise role of these lncRNAs in response to insulin is still not elucidated. However, the data suggest that metabolic regulation might be controlled by a single or a group of lncRNA and they could have a role in the pathogenesis of insulin-related pathologies.

Deep sequencing analysis followed by gain and loss of function approaches found that the circRNA was protective in metabolic syndrome patients [109]. In this study, the circRNF111 molecules were found to interact with miR-143 and inhibit the mediated insulin resistance caused by miR-143 overexpression. In another study, the group also identified IGF2R as a direct target for miR-143 [107]. These data agree with what was previously reported by Jordan and his group regarding miR-143 as discussed earlier [108]. Considering the findings from these recent studies, it is evident that a holistic approach should be followed when investigating metabolic drivers, ncRNA molecules, and their roles in the development of NAFLD.

## 6. Concluding Remarks

This review highlights the functional complexity of ncRNAs in NAFLD. The multitude of changes in the expression levels of different ncRNAs along with their target effectors may highlight the relevance of these ncRNAs in the different stages of NAFLD. Patients with NAFLD show distinct profiles of ncRNAs. The changes in the expression and/or localization of ncRNAs function as liver activators and contribute to the progression of NAFLD into NASH. Dysregulated ncRNAs were shown to upregulate lipotoxicity, oxidative stress fibrosis, inflammation, and metabolic dysfunction in NAFLD. Current treatment options involve medications that are effective only at improving disease progression. Weight loss achieved through adjusting lifestyle and diet is associated with improved steatosis and fibrosis. Drugs used for type 2 diabetes such as Vitamin E and pioglitazone and GLP-1 agonists were also shown to be effective in controlling NAFLD/NASH symptoms. Having more information about the NAFLD-regulated ncRNAs, and a comprehensive understanding of their precise roles in the disease would facilitate the development of highly effective targeted therapy. Despite the exponential increase in the field, the gap between the utility of ncRNA in NAFLD/NASH prevention and treatment is still lacking. Understanding the regulatory pathways involving ncRNA in NAFLD initiation and progression is challenging since other factors are implicated in the disease. To date, no ncRNA has been introduced to the scientific community as a novel agent for the treatment of NAFLD/NASH. Yet, with rapid technological breakthroughs, we are expecting novel insights that will further elucidate how NAFLD/NASH should be diagnosed, treated, and managed.

## Figures and Tables

**Figure 1 ncrna-10-00010-f001:**
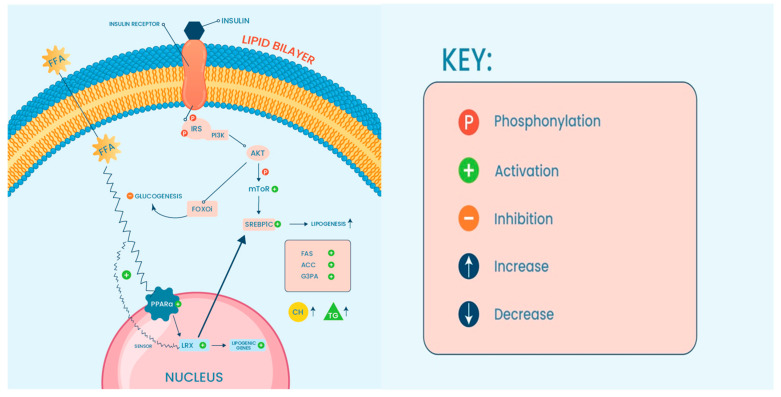
Key signaling pathway deregulation seen in NAFLD (see text for details).

**Figure 2 ncrna-10-00010-f002:**
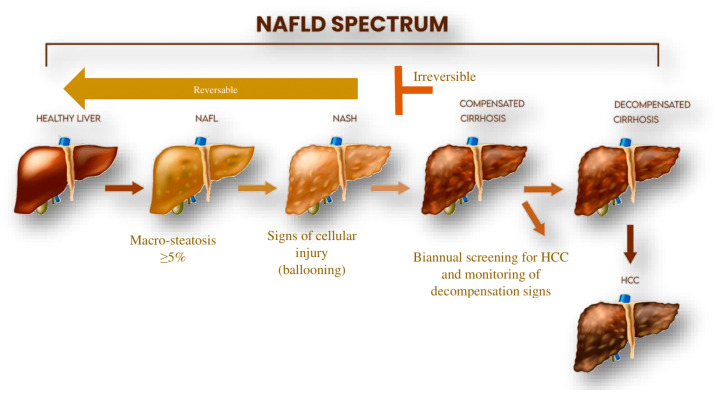
Non-alcaholic Fatty Liver progression. NAFL: non-alcoholic fatty liver, NASH: non-alcoholic steatohepatitis, HCC: hepatocellular carcinoma (see text for details).

**Figure 3 ncrna-10-00010-f003:**
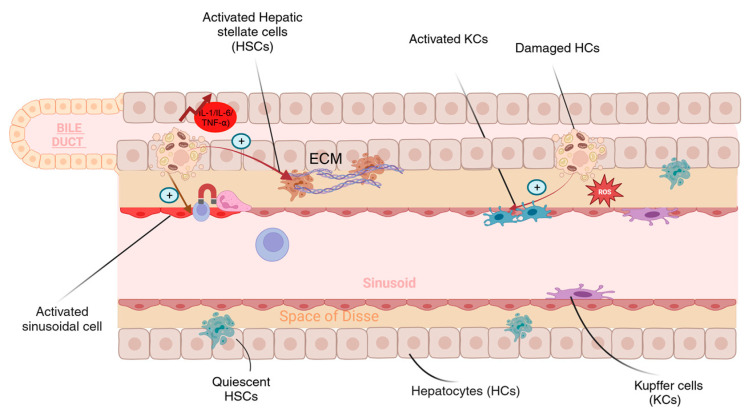
Due to dietary overload, injured hepatocytes (HCs) will induce the production of ROS and inflammatory cytokines, caused by hepatocyte organelle dysfunction. This will induce the release of endogenous signals that will enhance Kupffer cells (KCs) polarization. The prolonged release of pro-inflammatory mediators (IL-1, IL-6, and TNF-α) will subsequently activate HSCs and LSECs. HSCs are responsible for fibrogenic pathway activation while LSECs provide a platform for adhesion of blood-derived immune cells. The LSECs will also enhance the polarization of KCs (Created with BioRender.com).

**Figure 4 ncrna-10-00010-f004:**
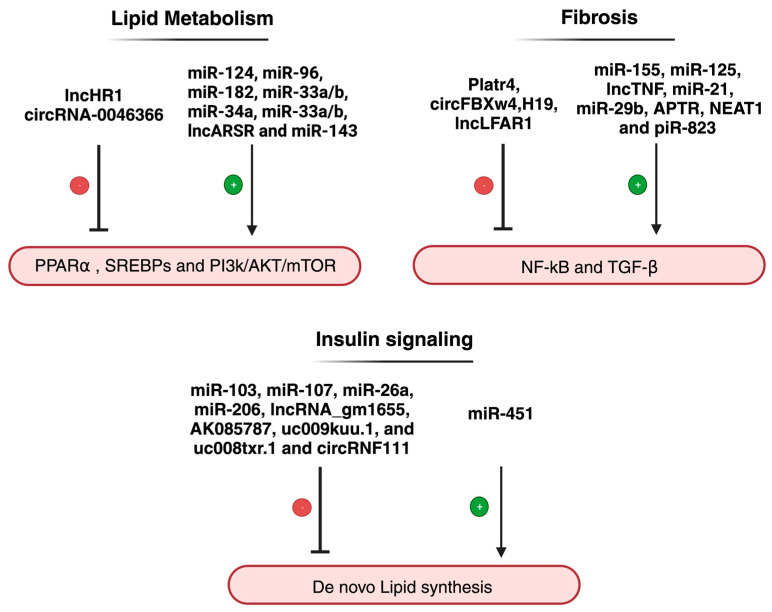
Regulation of major signaling pathways by ncRNA in NAFLD/NASH as discussed in the text (Created with BioRender.com).

**Table 1 ncrna-10-00010-t001:** List of ncRNAs involved in NAFLD.

Non-Coding RNA	Expression	Primary Function	Reference
miR-96-124-182-183	Upregulated	These ncRNAs modulate lipid synthesis through SREBP-1c	[89,90]
miR-206	Downregulated		[91]
Gm16551	Downregulated		[92]
lncHR1	Downregulated		[93]
miR-122-3075-21-192-5p	Upregulated	These ncRNAs are involved in the stimulation or inhibition of HCs and HC-related injuries	[46,49,54,56,57,58,74,77,94]
piR-823	Upregulated		[85]
circBNC2	Upregulated		[78]
circ_0071410			[77]
circPI4KB			[74]
circRNA_0046367	Downregulated		[72]
miR-26	Downregulated	miR-26 expression depends on LXRs and has a protective role	[95]
miR-34a	Upregulated	Both ncRNAs were found to be associated with the regulation of hepatic fat content through PPARα	[96]
circRNA_0046366	Downregulated		[96]
miR-451	Downregulated	Involved regulation of glucose homeostasis	[97]
miR-103/miR-107	Upregulated		[98]
LncARSR	Upregulated	The expression of LncARSR correlates with the activated state of PI3K/AKT/mTOR pathway	[99]
miR-155	Upregulated	These ncRNASs have an influence on the induction of inflammation via NF-κB pathway	[100,101]
Platr4	Upregulated		[102]
LncTNF	Upregulated		[103]
miR-125	Upregulated	The expression of these ncRNAs contributes to the activation of pro-inflammatory mediators	[104]
circFBXW4	Downregulated		[14]
miR-29b	Downregulated	These ncRNAs are involved in fibrosis mediation through TGF-β1	[94]
H19	Downregulated		[65,105]
piR-823	Upregulated		[85]
miR-26a-143-145	Upregulated	These ncRNAs have been found to be modulators in insulin sensitivity	[106,107,108]
AK085787-uc009kuu.1-uc008txr.1	Upregulated		[92]
CircRNF111-circRNA_0001805	Downregulated		[73,109]
CircRNA SCAR	Upregulated	This ncRNA may play a role in activating liver fibroblasts	[71]

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
