# Peer review of "Investigating the Role of Non-Coding RNA in Non-Alcoholic Fatty Liver Disease"

_ncrna, 2024, doi:10.3390/ncrna10010010_

Round 1

Reviewer 1 Report

Comments and Suggestions for Authors

The author has complied a good information document for non-coding RNAs in NAFLD.

Could the author please elabore the novelty of this review as there are multiple reviews published with the same theme with following link:

https://www.sciencedirect.com/science/article/abs/pii/S0065242322000439?via%3Dihub

https://academic.oup.com/hmg/article/31/R1/R4/6567452

https://www.mdpi.com/2218-273X/13/3/560

https://onlinelibrary.wiley.com/doi/full/10.1002/jcla.24943

https://www.ncbi.nlm.nih.gov/pmc/articles/PMC7593715/

Please put references in line 156-165 with paragraph "In simple steatosis.."

Please put reference for line number 169-170 with line "Prolonged activation of the innate immune response..."

Please put some online resources such as databases to look for any non-coding RNAs for researchers.

Author Response

Reviewer #1: We thank the reviewer for the positive comments. We made the suggested changes in the revised version of the manuscript.

Comment 1: The author has complied a good information document for non-coding RNAs in NAFLD. Could the author please elabore the novelty of this review as there are multiple reviews published with the same theme with following link:

https://www.sciencedirect.com/science/article/abs/pii/S0065242322000439?via%3Dihub

https://academic.oup.com/hmg/article/31/R1/R4/6567452

https://www.mdpi.com/2218-273X/13/3/560 https://onlinelibrary.wiley.com/doi/full/10.1002/jcla.24943

https://www.ncbi.nlm.nih.gov/pmc/articles/PMC7593715/

PI response: We carefully reviewed the literature published on the same theme and we think the novelty of our review article lies in several aspects. Firstly, our review provides a comprehensive view of the involvement of most studies non-coding RNA including long noncoding RNA (lncRNA), circular RNA (circRNA), piwi RNA (piRNA) and microRNA (miRNA) in NAFLD, providing are linked to metabolic and fibrogenic pathways dysfunction. We believe this review article provides a more understanding of the subject than what is available in the published reviews. We found that previous reviews have covered a wild disciplinary aspect of non-alcoholic fatty acid diseases and the role of non-coding RNA in general. Secondly, previous review did not cover current work in about Piwi RNA involvement in the development pf NAFLD which was covered in our review. Furthermore, a number of published articles are systematic reviews that included only long noncoding RNA and circular RNA, where in our review we have qualitatively summarized the involvement of non-coding RNA, and provided a comprehensive understanding of the topic, helping in identifying the research gaps, and shedding the lights on complexity of the disease phenomena. 

Comment 2: Please put references in line 156-165 with paragraph "In simple steatosis.."

PI response: As suggested by the reviewer, we have included the references with the numbers 27-29 in line 166.

Comment 3: Please put reference for line number 169-170 with line "Prolonged activation of the innate immune response..."

PI response: As suggested by the reviewer, we have included reference number 32 in line 188.

Comment 4: Please put some online resources such as databases to look for any non-coding RNAs for researchers.

PI response: This is a great suggestion by the reviewer. 

Publicly available RNA database:

RNAcentral: The non-coding RNA sequence database. Information about this database can be found at https://doi.org/10.1093/nar/gkw1008

ncRNA Databases. Information about this database can be found at https://doi.org/10.1093/nar/gkl994

GeneCaRNA - The Human ncRNA Database (genecards.org). Information about this database can be found https://doi.org/10.1016/j.jmb.2021.166913

Reviewer 2 Report

Comments and Suggestions for Authors

1.More informations about the function of lncRNAs were recommended to supply  in the abstract section.

2.HSCs can also interact with macrophages, this section should be depicted more (in the 2.1)

3.lncRNA NEAT1 plays an important role in regulating NAFLD progression, but it isn't mentioned it this review.

4. In the part of fibrosis,  some miRNAs associated with TGF-β pathway and the level of them reduction, but only these with elevated levels  were mentioned. More descriptions about circRNA related to fibrosis, such as circDIDO1, circCDK13 and so on should be added in the paper.

Author Response

Reviewer #2: We thank the reviewer for the comments. We agree with provided comments and we made the suggested changes in the revised version of the manuscript.

Comment 1: More informations about the function of lncRNAs were recommended to supply in the abstract section.

-PI response. Thank you. Information regarding lncRNA have been expanded in the review.

Comment 2: HSCs can also interact with macrophages, this section should be depicted more (in the 2.1)

PI response: Thank you for bringing this up. HSCs interaction is added to section 2.1 paragraph 2 line 146-155

Comment 3: lncRNA NEAT1 plays an important role in regulating NAFLD progression, but it isn't mentioned it this review.

PI response: Thanks for bringing this up. We have included information of NEAT1 from line 352 to 364 and from line 623-628. We have updated the reference list accordingly.

Comment 4:  In the part of fibrosis, some miRNAs associated with TGF-βpathway and the level of them reduction, but only these with elevated levels were mentioned. More descriptions about circRNA related to fibrosis, such as circDIDO1, circCDK13 and so on should be added in the paper.

PI response: Thank you for this comment. This has been added in line 394-404

Reviewer 3 Report

Comments and Suggestions for Authors

The present manuscript has reviewed the molecular biological contribution of non-coding RNA in non-alcoholic fatty liver disease (NAFLD). The reviewer considers that the present manuscript is well-written and well-constructed. The reviewer would like to indicate some requests as follows.

1. In section 5 and 6, the authors described the molecular mechanism that the ncRNAs are related to various biological events which contribute to the pathological progression of NAFLD.  The reviewer considers that those contents are very important points in the present manuscript. So, the reviewer would request to make one or two figures to display the molecular biological relationship between ncRNAs and the progression of NAFLD (the up and down-regulation of ncRNAs, the key functional proteins, the signaling pathways). 

2. In Table 1, the reviewer considers the reference numbers had better be added to each ncRNA.

3. The proofreading should be performed again. There are some unabbreviated words in the main text.

Comments on the Quality of English Language

The minor proofreading should be performed.

Author Response

Reviewer #3: We thank the reviewer for the feedback. We made the suggested changes in the revised version of the manuscript.

Comment 1: In section 5 and 6, the authors described the molecular mechanism that the ncRNAs are related to various biological events which contribute to the pathological progression of NAFLD. The reviewer considers that those contents are very important points in the present manuscript. So, the reviewer would request to make one or two figures to display the molecular biological relationship between ncRNAs and the progression of NAFLD (the up and down-regulation of ncRNAs, the key functional proteins, signaling pathways).

PI response: This is a great comment. We have included Figure 3 and Figure 4 to the manuscript.

Comment 2: In Table 1, the reviewer considers the reference numbers had better be added to each ncRNA.

PI response: As suggested by the reviewer we added the reference numbers to each ncRNA in the table.

Comment 3: The proofreading should be performed again. There are some unabbreviated words in the main text.

PI response: We thank the reviewer for the comment. We have conducted a carful proofreading which strengthen our review article.

Reviewer 4 Report

Comments and Suggestions for Authors

Non-coding RNAs (ncRNAs) play crucial roles in regulating cellular processes by globally influencing gene expression. This review explores the involvement of ncRNAs in the pathogenesis of non-alcoholic fatty liver disease (NAFLD), discussing its clinical features, treatment options, and the dysregulated signaling pathways that interact with ncRNAs. While the work may appear useful for a scientific audience, it could benefit from addressing comments related to facts, clarity and organization to avoid potential confusion among readers.

1.     A critical aspect that demands meticulous consideration is the differentiation of this review from previously published works, specifically those conducted by Huang et al in 2019 (https://doi.org/10.1155%2F2019%2F8690592), Khalifa et al in 2020 (https://doi.org/10.1155%2F2019%2F8690592), Rusu et al in 2022 (https://doi.org/10.3390/ijms232012370), and Shi et al in 2023 (https://doi.org/10.1002/jcla.24943). This concern underscores the necessity to articulate how the current work distinguishes itself in terms of objectives, , scope, and findings when compared to the aforementioned publications. A meticulous examination of these distinctions is paramount to elucidating the unique contributions and novel insights offered by the present review in the landscape of non-coding RNA and intercellular communication within liver cell populations.

2.     Consider adding figures or diagrams to visually represent the different cell types and their interactions, especially for section 2.1. This can aid in comprehension and make the text more engaging.

3.     The section on "2.1 Pathogenesis and clinical aspects of NAFLD/NASH" is quite detailed but may benefit from further elaboration on specific mechanisms or recent research findings related to NAFLD/NASH.

4.     Ensure consistency in the use of terminology. For instance, there is mention of "hepatocytes" and "liver cells" - make sure these terms are used consistently. Avoid unnecessary repetition, such as repeatedly stating that hepatocytes are the primary functional unit of the liver. Once established, subsequent references can simply use "hepatocytes."

5.     Section 2.2: Some information is repeated or could be condensed for brevity. For instance: (Line 200) "To date, there is no approved treatment for NAFLD/NASH, but a few drugs have been shown to be effective at improving the prognosis of the disease." Authors should consider rephrasing to avoid redundancy.

6.     Section 2.2: Include information on ongoing research, clinical trials, or emerging treatment strategies for NAFLD/NASH to make the review more comprehensive. Consider adding a brief mention of promising treatments or therapeutic approaches currently under investigation.

7.      Was there any ‘section 3’? I was able to find Sections 2 and 4, but Section 3 appears to be missing (or could it be a typing error?). Additionally, please conduct a thorough proofread of the text to rectify any grammatical errors and ensure a refined final review.

8.     Change miR-34 as miR-34a in Table 1 and in lines between 407 and 411 on page 10

9.     Several miRNAs implicated in NAFLD is not discuused in the review. This includes miR-33 a/b. And the role of lncRNAs such as Alu-Mediated p21 Transcriptional Regulator (APTR), Homeobox (HOX) Transcript Antisense RNA (HOTAIR), Nuclear Enriched Abundant Transcript1(NEAT1) etc

10.  Ensure consistency in the use of terminology. For example, consider using either "ncRNA" or "non-coding RNA" consistently throughout the text.

11.  Mentioning any limitations of the reviewed studies or areas requiring further research could add depth to the critical analysis.

12.  Line 529: In the sentence "Recently, it was suggested to rename NAFLD to metabolic-associated non-alcoholic fatty liver disease (MAFLD) [104].", consider providing a brief rationale or key points behind the suggested renaming for better context.

13.  Line 594: Add specific references to support general statements, such as "Recent studies have shown that NAFLD is a multisystem disease affecting various body systems." Providing specific examples or findings from recent studies would add depth to this statement.

14.  Line 595: In the sentence "To date, no approved treatment has been identified, and current treatment options involve medications that are effective only at the progression of the disease," authors should consider specifying which medications are being referred to and their limitations. This would provide a clearer understanding of the current state of treatment options.

15.  Line 598: what does NFLD stand for? Authors should ensure consistent terminology. For example, consider using either "NAFLD" or "NFLD" consistently throughout the text.

16.  Page 14: Section 7; The conclusion could be strengthened by summarizing the key findings and implications presented in the review. Consider rephrasing to emphasize the importance of continued research in understanding the roles of ncRNAs in NAFLD and the potential for targeted therapies.

Comments on the Quality of English Language

Please refer above.

Author Response

Reviewer #4: We thank the reviewer for providing a very constructive comments. We agree with provided feedback and we made the suggested changes in the revised version of the manuscript.

Comment 1: A critical aspect that demands meticulous consideration is the differentiation of this review from previously published works, specifically those conducted by Huang et al in 2019(https://doi.org/10.1155%2F2019%2F8690592), Khalifa et al in2020 (https://doi.org/10.1155%2F2019%2F8690592), Rusu et al in2022 (https://doi.org/10.3390/ijms232012370), and Shi et al in 2023(https://doi.org/10.1002/jcla.24943). This concern underscores the necessity to articulate how the current work distinguishes itself in terms of objectives, , scope, and findings when compared to the aforementioned publications. A meticulous examination of these distinctions is paramount to elucidating the unique contributions and novel insights offered by the present review in the landscape of non-coding RNA and intercellular communication within liver cell populations.

PI response: We completely agree with the Reviewer 1 and 4 who brought up this concerns and we have addressed this in detail in Reviewer 1 comment 1.

Comment 2: Consider adding figures or diagrams to visually represent the different cell types and their interactions, especially for section 2.1. This can aid in comprehension and make the text more engaging.

PI response: As suggested by the reviewer we added two additional figures to the manuscript: Figure 3 and Figure 4.

Comment 3: The section on "2.1 Pathogenesis and clinical aspects of NAFLD/NASH" is quite detailed but may benefit from further elaboration on specific mechanisms or recent research findings related to NAFLD/NASH.

PI response: We thank the reviewer for this comment. We added explanation of the current findings in 167 to 177.

Comment 4: Ensure consistency in the use of terminology. For instance, there is mention of "hepatocytes" and "liver cells" - make sure these terms are used consistently. Avoid unnecessary repetition, such as repeatedly stating that hepatocytes are the primary functional unit of the liver. Once established, subsequent references can simply use "hepatocytes."

PI response: Thank you for bringing this issue to our attention. We have gone through the document and made adjustments as necessary to ensure the terms are used consistently, including for “hepatocytes” and “liver cells”.

Comment 5: Section 2.2: Some information is repeated or could be condensed for brevity. For instance: (Line 200) "To date, there is no approved treatment for NAFLD/NASH, but a few drugs have been shown to be effective at improving the prognosis of the disease. “Authors should consider rephrasing to avoid redundancy.

PI response: Thank you. We have gone through and tried to make sure that we do not repeat information and to make sure that sentences are written succinctly. We also tried to make sure that abbreviations for the types of RNA were used after each type had been spelled out the first time.

Comment 6: Section 2.2: Include information on ongoing research, clinical trials, or emerging treatment strategies for NAFLD/NASH to make the review more comprehensive. Consider adding a brief mention of promising treatments or therapeutic approaches currently under investigation.

PI response: As reviewer suggested, we have added some information of the current promising treatment and clinical trial in 223-227 and 245-249

Comment 7: Change miR-34 as miR-34a in Table 1 and in lines between 407 and 411 on page 10

PI response: As reviewer suggested, we changed miR-34 as miR-34a in Table 1 and we made sure it is constant in the text.

Comment 8: Was there any ‘section 3’? I was able to find Sections 2 and 4, but Section 3 appears to be missing (or could it be a typing error?). Additionally, please conduct a thorough proofread of the text to rectify any grammatical errors and ensure a refined final review.

PI response: We thank the reviewer for the comment. We corrected the section numbering through the manuscript. We also conducted a thoroughly proofreading and corrected grammatical errors. 

Comment 9: Several miRNAs implicated in NAFLD is not discussed in review. This includes miR-33 a/b. And the role of lncRNAs such as Alu-Mediated p21 Transcriptional Regulator (APTR), Homeobox(HOX) Transcript Antisense RNA (HOTAIR), Nuclear Enriched Abundant Transcript1(NEAT1) etc

PI response: Thanks for your comment. We have discussed these ncRNA in the review. New paragraphs been and highlighted on the text under the following sections: 3.2, 3.3 and 4.2,

Comment 10: Ensure consistency in the use of terminology. For example, consider using either "ncRNA" or "non-coding RNA" consistently throughout the text.

PI response: As suggested, we have unified the terminology throughout the manuscript.

Comment 11: Mentioning any limitations of the reviewed studies or are as requiring further research could add depth to the critical analysis.

PI response: We have considered adding study limitations in the conclusion section.

Comment 12: Line 529: In the sentence "Recently, it was suggested to rename NAFLD to metabolic-associated non-alcoholic fatty liver disease (MAFLD) [104].", consider providing a brief rationale or key points behind the suggested renaming for better context.

PI response: Thank you for your suggestion. We have added ‘MAFLD reflects the pathological factors that contribute to the development of fatty liver and give a comprehensive understanding of the disease progression and management’ in line 560-562.

Comment 13: Line 594: Add specific references to support general statements, such as "Recent studies have shown that NAFLD is a multisystem disease affecting various body systems." Providing specific examples or findings from recent studies would add depth to this statement.

PI response: We thank the reviewer for the comment. We have revised the conclusion as addressed in Reviewer 4 comment 16.

Comment 14: Line 595: In the sentence "To date, no approved treatment has been identified, and current treatment options involve medications that are effective only at the progression of the disease," authors should consider specifying which medications are being referred to and their limitations. This would provide a clearer understanding of the current state of treatment options.

PI response: As suggested by the reviewer, we have revised this line and specified treatments options. This section has been modified: “Currently, there is no approved treatment has been identified for NAFLD/ NASH, and current treatment options involve medications that are effective only at the progression of the disease. Weight loss achieved through adjusting lifestyle and diet is associated with improved steatosis and fibrosis. Drugs used for type 2 diabetes such as Vitamin E and pioglitazone, and GLP-1 agonists we also shown to be effective in controlling NAFLD/NASH symptoms.”

Comment 15: Line 598: what does NFLD stand for? Authors should ensure consistent terminology. For example, consider using either "NAFLD" or "NFLD" consistently throughout the text.

PI response: As suggested by the reviewer, we ensured the consistency of terminology and marinated NAFLD throughout the manuscript.  

Comment 16: Page 14: Section 7; The conclusion could be strengthened by summarizing the key findings and implications presented in the review. Consider rephrasing to emphasize the importance of continued research in understanding the roles of ncRNAs in NAFLD and the potential for targeted therapies.

PI response: We thank the reviewer for the comment. We have revised the conclusion as follow: “This review highlights the functional complexity of ncRNAs in NAFLD. The multitude of changes in the expression levels of different ncRNAs along with their target effectors may highlight the relevance of these ncRNAs in the different stages of NAFLD. Patients with NAFLD show distinct profiles of ncRNAs. The changes in the expression and/or localization of ncRNAs function as liver activators and contribute to the progression of NAFLD into NASH. Dysregulated ncRNAs were shown to upregulate lipotoxicity, oxidative stress fibrosis, inflammation, and metabolic dysfunction in NAFLD. Current treatment options involve medications that are effective only at improving disease progression. Weight loss achieved through adjusting lifestyle and diet is associated with improved steatosis and fibrosis. Drugs used for type 2 diabetes such as Vitamin E and pioglitazone and GLP-1 agonists were also shown to be effective in controlling NAFLD/NASH symptoms. Having more information about the NAFLD-regulated ncRNAs, and a comprehensive understanding of their precise roles in the disease would facilitate the development of highly effective targeted therapy. Despite the exponential increase in the field, the gap between the utility of ncRNA in NAFLD/NASH prevention and treatment is still lacking. Understanding the regulatory pathways involving ncRNA in NAFLD initiation and progression is challenging since other factors are implicated in the disease. To date, no ncRNA has been introduced to the scientific community as a novel agent for the treatment of NAFLD/NASH. Yet, with rapid technological breakthroughs, we are expecting novel insights that will further elucidate how NAFLD NASH should be diagnosed, treated, and managed.”

Round 2

Reviewer 1 Report

Comments and Suggestions for Authors

No comments.

Reviewer 2 Report

Comments and Suggestions for Authors

Accept in present form.

Reviewer 4 Report

Comments and Suggestions for Authors

Upon reviewing the authors' detailed responses, it is evident that the suggested corrections have been duly attended to.

Comments on the Quality of English Language

N.A.